# Phase Transitions and Hub Vulnerability in Networked Truth Recovery under Misinformation

## Abstract

This paper examines how networked agents recover ground truth when initial knowledge is incomplete and contaminated by misinformation. We ask: (1) under what conditions can networks reconstruct the correct knowledge base, (2) how does centrality affect recovery across informational environments, and (3) what structural vulnerabilities emerge when misinformation dominates? We develop a multi-agent simulation where agents begin with a mix of true and false facts, exchange information with neighbors, and update beliefs using a redundancy-based scoring rule with contradiction resolution. The system aims to reconstruct the full ground-truth knowledge base. Experiments on the ca-GrQc collaboration network reveal abrupt bandwidth thresholds: below a critical share budget (the maximum number of facts an agent can transmit per round), convergence never occurs; above it, recovery typically completes within 1–3 rounds and yields complete accuracy across agents that converge. Information quality defines three regimes. In clean environments (truth $\geq 70\%$), centrality strongly predicts success. At a boundary near 50% truth, centrality effects collapse ($\rho \approx 0.2$ or less). In polluted environments (truth $\leq 40\%$), central nodes amplify errors, producing a *hub vulnerability paradox*. These findings identify structural weaknesses in information systems and show that resilience depends jointly on communication capacity, network topology, and information quality.

## 1 Introduction

The challenge of forming accurate collective beliefs from distributed, partial, and often noisy information is a fundamental problem in social systems. Individuals rarely possess a complete picture of reality; instead, they receive fragments of information through their social connections, some of which may be incorrect or deliberately misleading. This process is central to how societies establish norms, make collective decisions, and construct a shared understanding of the world. With the growth of online social networks, the scale and speed of this process have increased dramatically, making the study of collective belief formation more urgent than ever. Classic theories of diffusion and social learning describe how individuals aggregate information, but they often assume either reliable signals or gradual, continuous adjustment—assumptions that break down in environments saturated with misinformation.

To study these dynamics, we propose a minimal multi-agent model in which agents begin with a mixture of true and false facts, exchange information with their neighbors, and update beliefs using a redundancy-based heuristic with contradiction resolution. This heuristic is deliberately simple: agents count repetitions of facts without assessing credibility, coherence, or source reliability. Such a rule is intentionally conservative, since it makes recovery more difficult than under richer cognitive strategies. Any successful recovery observed under this baseline therefore represents a lower bound on what more sophisticated reasoning agents might achieve.

Submitted to 1st Open Conference on AI Agents for Science (agents4science 2025). Do not distribute.

Recent advances in large language models (LLMs) provide an additional motivation. LLMs are trained on web-scale corpora that contain both reliable information and misinformation, and their tendency to confidently generate plausible falsehoods parallels how individual agents propagate errors in social networks. As LLMs are increasingly deployed in multi-agent settings—interacting with one another to solve problems or negotiate—they mirror the collective dynamics we study. To illustrate this connection, we include small demonstration runs with LLaMA 3.0 agents, while the main results rely on large-scale symbolic simulations.

This framework enables systematic study of how collective truth recovery depends on three key factors: information quality, network topology, and communication bandwidth. We focus on three questions: (1) under what conditions can networks reconstruct the correct knowledge base; (2) how does centrality affect recovery across environments; and (3) what structural vulnerabilities emerge when misinformation dominates?

**Contributions.** This paper makes three contributions: (1) We formalize a simulation model of distributed truth recovery under misinformation, providing a transparent baseline where agents operate with minimal cognitive assumptions. (2) Using symbolic experiments on the ca-GrQc collaboration network, we show bandwidth-dependent thresholds and identify three regimes—clean, boundary, and polluted—with distinct centrality–performance relationships. (3) We uncover and formalize the *hub vulnerability paradox*: in misinformation-rich settings, centrality becomes harmful, as highly connected agents amplify falsehood rather than truth, reversing classical diffusion expectations. This finding reveals structural fragilities in both human networks and highlights the need to examine similar risks in emerging multi-agent LLM systems.

## 2   Related Work

Traditional social science has often relied on surveys and controlled experiments to investigate individual and collective behavior. While these methods yield authentic data, they are expensive, hard to scale, and sometimes raise ethical concerns [15, 18, 2, 22]. The rise of large language models (LLMs) offers a promising alternative, as they demonstrate capabilities in reasoning, planning, and role-playing that allow them to simulate human responses with high fidelity [32, 19, 1, 6]. This quality, sometimes referred to as algorithmic fidelity, makes LLM-based agents especially useful for reproducing human behaviors in controlled settings.

Current research in LLM-based social simulation is commonly organized into three tiers. **Individual simulation** focuses on replicating distinct people or demographic groups, modeling attributes such as personality, social roles, or even well-known public figures [7, 31, 3]. **Scenario simulation** brings together multiple agents within structured contexts to solve domain-specific problems—for example, collaborative tasks in software engineering, healthcare, or legal decision-making [30, 11, 4, 12, 16, 26, 14]. These studies highlight the potential for agents with complementary skills to achieve collective intelligence, mirroring cooperative dynamics in human groups. Finally, **society-level simulation** examines emergent macro-level dynamics such as opinion formation, polarization, and economic behavior, arising from interactions among large numbers of heterogeneous agents [27, 17, 10, 9, 23, 21, 20, 8, 25, 24].

These three layers are interrelated: individual simulations provide the foundation for task-based scenarios, which in turn scale into society-level models. Together, they form a continuum from fine-grained modeling of single agents to the study of collective processes in large populations [13, 5]. Prior surveys have mostly concentrated on isolated aspects such as agent architectures or single-agent role-playing [28, 29, 31, 13, 33]. What distinguishes recent work is its effort to unify research at all three levels—individual, scenario, and societal—providing a more systematic framework for studying how distributed agents, whether human or artificial, can overcome noisy or incomplete information to converge on shared understandings.

## 3   Methodology

We developed a multi-agent simulation platform to study how networked agents reconstruct a shared reality from distributed and noisy information. Our goal was to create a minimal yet interpretable setting where the conditions for collective truth recovery could be isolated and systematically tested.

## 3.1 Information Environment

We define a finite set of $N$ elementary propositions, $\mathcal{P} = \{p_1, p_2, \ldots, p_N\}$. The complete knowledge universe is

$$\mathcal{U} = \{p_i, \neg p_i \mid p_i \in \mathcal{P}\},$$

which has size $|\mathcal{U}| = 2N$. Each element $f \in \mathcal{U}$ is treated as a candidate fact. From this universe we construct a latent ground-truth knowledge base (KB), $\mathcal{KB} \subset \mathcal{U}$, by sampling exactly one fact for each proposition. Thus, $|\mathcal{KB}| = N$, and for every $p_i \in \mathcal{P}$, either $p_i$ or $\neg p_i$ is in $\mathcal{KB}$. This ensures completeness and consistency: there is a definitive truth value for every proposition, but agents do not initially know which version is correct. The complement $\mathcal{U} \setminus \mathcal{KB}$ is the set of false facts.

In the main experiments we set $N = 20$, creating a universe of 40 possible facts. Larger $N$ values require more rounds of exchange for agents to reconstruct the KB; with our 30-round horizon, recovery becomes infeasible in practice. Pilot tests with $N = 30$ and $N = 40$ confirmed qualitatively similar results, suggesting that the findings are not sensitive to the exact size of the knowledge base but are easier to study with $N = 20$.

## 3.2 Agents and Network

Agents are situated in a social network represented as an undirected graph $G = (V, E)$. Each node $a_i \in V$ is an agent, and edges represent communication links. The neighborhood of agent $a_i$ is denoted $N(i)$. The network constrains who can communicate with whom, which in turn shapes the collective ability to recover the truth.

At initialization ($t = 0$), each agent $a_i$ is assigned a knowledge set $\mathcal{K}_i^0$ composed of both true and false facts. Specifically, $k_{\text{true}}$ facts are sampled with replacement from $\mathcal{KB}$, and $k_{\text{false}}$ from $\mathcal{U} \setminus \mathcal{KB}$. Sampling with replacement means that an agent may hold duplicate items initially; duplicates are stored once but contribute to a higher starting score for that fact. This design guarantees that every agent begins with partial and noisy information, mimicking realistic conditions where individuals are exposed to both accurate and misleading claims. Agents know the parameters $N$, $k_{\text{true}}$, and $k_{\text{false}}$, but not the truth status of any fact.

**Symbolic vs. LLM instantiation.** The main experiments use symbolic agents that follow simple redundancy-based update rules. This makes it feasible to simulate thousands of nodes on the full ca-GrQc network, isolate structural effects, and obtain stable statistics across seeds. We also conducted smaller demonstration runs with LLaMA 3.0 agents. In these demos, each agent was instantiated as an LLM process, received neighbor facts as a prompt, and output an updated belief set in natural language. These runs were illustrative only; their purpose was to check whether the symbolic redundancy heuristic plausibly reflects behaviors observed in LLM interaction. The design of these demonstrations is described in the *Experimental Setup*, and their qualitative outcomes are discussed in the *Discussion*.

**Information exchange (selection policy).** At each step $t$, agent $a_i$ forms scores $s_i^t(f)$ for $f \in \mathcal{K}_i^t$ and transmits up to $B$ facts with the highest scores. Ties are broken deterministically by lexicographic order on identifiers; robustness checks with random tie-breaking yielded qualitatively similar results. Let $\mathcal{M}_i^t$ denote this transmitted set (*Top-B by score*). This rule reflects the intuition that agents prefer to share information they consider most reliable, subject to communication bandwidth limits.

**Belief updating.** Upon receiving messages, agent $a_i$ updates its knowledge as

$$\mathcal{K}_i^{t+1} = \mathcal{K}_i^t \cup \bigcup_{j \in N(i)} \mathcal{M}_j^t,$$

and increments scores

$$s_i^{t+1}(f) = s_i^t(f) + \sum_{j \in N(i)} \mathbb{I}\left(f \in \mathcal{M}_j^t\right),$$

with $s_i^0(f) = 1$ for $f \in \mathcal{K}_i^0$ and 0 otherwise. For evaluation, we compute:

$$\text{Precision}_i^t = \frac{|\mathcal{K}_i^t \cap \mathcal{KB}|}{|\mathcal{K}_i^t|}, \quad \text{Recall}_i^t = \frac{|\mathcal{K}_i^t \cap \mathcal{KB}|}{N}.$$

Exact recovery is defined as recall = 1.

**Contradiction resolution.** If both $p_i$ and $\neg p_i$ are present, the item with the higher score is retained and its negation removed. If the scores are equal, the agent retains neither (the *defer rule*) until a strict score advantage arises. This conservative policy avoids arbitrarily privileging one fact, though it may slow recovery.

**Termination.** The simulation ends when any agent reconstructs all $N$ true facts ("first valid proposer" rule) or when a maximum of 30 rounds is reached. Although runs stop at the first proposer, we evaluate precision, recall, and exact recovery across all agents at termination. Reported performance distributions therefore reflect the state of the entire population, not only the first proposer.

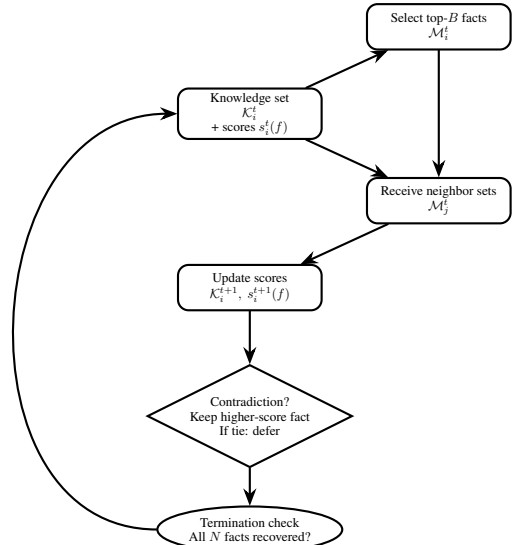

Figure 1: Belief update cycle for an agent $a_i$. Each round: (1) select top-$B$ facts by score, (2) transmit to neighbors, (3) receive neighbor messages, (4) update scores, (5) resolve contradictions, and (6) check for termination.

# 4 Experimental Setup

## 4.1 Network Dataset

We use the ca-GrQc collaboration graph from SNAP, with 4,158 nodes and 13,422 undirected edges. The graph has heterogeneous degree, clustering, and short paths typical of social systems. This size allows full-network simulations with stable statistics while keeping runtimes practical.

## 4.2 Initialization Regimes and Bandwidth

Each agent starts with a mix of true and false facts. We study five regimes,

$$(k_{\text{true}}, k_{\text{false}}) \in \{(7,1),\ (5,2),\ (4,3),\ (4,4),\ (3,5)\},$$

which correspond to truth ratios of 87.5%, 71.4%, 57.1%, 50.0%, and 37.5%. These settings cover clean ($\geq 70\%$), transition (50–60%), and polluted ($\leq 40\%$) environments. Pilot runs with nearby values produced the same qualitative behavior, so we focus on these five for tractability.

Communication is limited by a share budget $B$ (maximum facts transmitted per round). We test

$$B \in \{1, 2, 3, 4, 5, 6, 7, 8, 10, 12, 14, 20\},$$

with low $B$ stressing tight bandwidth, mid-range $B$ targeting the threshold region, and high $B$ probing whether extra capacity helps once past the threshold.

For each regime–$B$ condition we run three independent seeds (42, 43, 44), yielding 93 runs and over 150,000 agent-level records. A subset of regime–$B$ pairs is omitted to concentrate compute on the threshold region and representative extremes; the selection is shown in Table 1.

Table 1: Experimental design: five information regimes crossed with twelve bandwidth values. Checks mark evaluated cells. Each evaluated cell was run with three seeds (42, 43, 44). Not all pairs were run; we focus on values near thresholds and representative extremes.

| Regime | 1 | 2 | 3 | 4 | 5 | 6 | 7 | 8 | 10 | 12 | 14 | 20 |
|---|---|---|---|---|---|---|---|---|---|---|---|---|
| (7,1) 87.5% | ✓ | ✓ | | | ✓ | ✓ | ✓ | ✓ | | ✓ | | |
| (5,2) 71.4% | ✓ | ✓ | ✓ | ✓ | ✓ | ✓ | ✓ | ✓ | ✓ | ✓ | | |
| (4,3) 57.1% | | | | | | ✓ | ✓ | ✓ | | | | |
| (4,4) 50.0% | | | | | ✓ | ✓ | ✓ | ✓ | | | | |
| (3,5) 37.5% | | | | ✓ | ✓ | ✓ | | ✓ | | ✓ | ✓ | ✓ |

## 4.3 LLaMA Demonstrations

In addition to the symbolic simulations, we ran small-scale demonstrations with LLaMA 3.0 agents to check whether the symbolic redundancy heuristic plausibly reflects behaviors in real language models. These runs used 50–100 agents drawn from connected subgraphs of the ca-GrQc network and were simulated for up to 10 rounds. Each subgraph preserved local degree and clustering structure while keeping the size small enough for interactive LLaMA calls. At each round, the agent received its current knowledge set $\mathcal{K}_i^t$ and the transmitted items from its neighbors as a text prompt. The prompt was formatted as a short instruction block, for example:

> "You are Agent #17. Here are the assertions you currently believe: [list of $\mathcal{K}_i^t$]. Here are the assertions just received from your neighbors: [list of $\mathcal{M}_j^t$]. Update your belief set. Keep assertions that are repeated, drop those contradicted by stronger evidence, and output your top 5 most reliable assertions."

The model's response was parsed to extract the updated set $\mathcal{K}_i^{t+1}$ and associated scores. To keep the demonstration close to the symbolic rules, the prompt included explicit instructions about repetition counting and contradiction resolution, and we limited output length to match the symbolic share budget $B$.

## 4.4 Outcome Measures and Criteria

Agent accuracy is the fraction of items in $\mathcal{K}_i^t$ that match $\mathcal{KB}$ at evaluation time. A run *converges* if at least one agent reconstructs all $N$ true facts before the 30-round cap (first-valid-proposer rule). For each condition we report: convergence (yes/no), median and range of rounds-to-convergence across successful seeds, and the distribution of agent accuracies at termination. We also compute Spearman correlations ($\rho$) between agent performance and four centrality measures (degree, betweenness, closeness, eigenvector), with bootstrap 95% CIs (5,000 resamples within run, then pooled by regime).

# 5 Results

We report findings from 93 experimental runs (5 regimes × tested bandwidths × 3 seeds), producing over 150,000 agent-level performance records. Results are organized around three discoveries: (i) threshold effects in recovery, (ii) reversal of centrality effects, and (iii) convergence timing.

## 5.1 Phase Transitions in Truth Recovery

Truth recovery depended critically on communication bandwidth $B$ and the quality of initial information. Figure 2 summarizes convergence outcomes. Clean regimes (87.5% and 71.4%) display sharp thresholds: configuration $(7, 1)$ converges reliably once $B \geq 7$, and $(5, 2)$ converges once $B \geq 6$. The boundary regime $(4, 3)$ shows partial convergence at $B = 7$ and achieves full convergence by $B = 8$. In contrast, the 50% boundary $(4, 4)$ and polluted $(3, 5)$ regimes fail to converge even under very high bandwidths ($B = 14, 20$).

We define *convergence* as at least one agent reconstructing the full KB before the 30-round horizon. A regime–$B$ condition is classified as *full convergence* if all three seeds converge, *partial convergence* if two of three converge, and *no convergence* if one or zero converge. This convention captures

sensitivity to random initialization rather than noise in the update rules. Once above the regime-specific threshold ($B \approx 6$–$8$), additional bandwidth did not reduce convergence time or improve accuracy. The transitions are discontinuous: the system shifts from non-recovery to complete recovery without intermediate states, a phenomenon we describe as a *phase transition* in a qualitative sense (sharp tipping behavior) rather than in the strict statistical physics sense of estimating critical exponents.

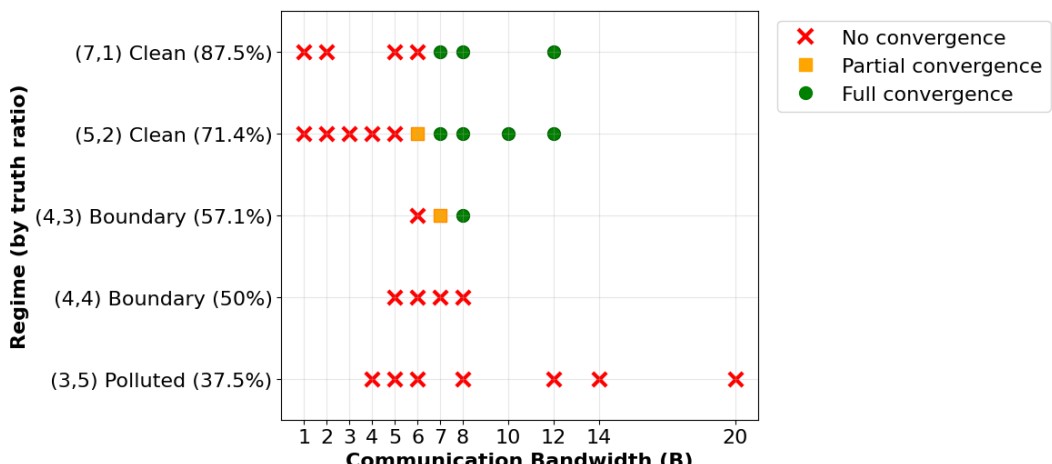

Figure 2: **Convergence outcomes across bandwidth and information regimes.** X-axis: bandwidth $B$; Y-axis: regime by truth ratio. Markers: red X = no convergence, orange square = partial convergence (2/3 seeds), green circle = full convergence (3/3 seeds). Clean regimes converge once $B$ reaches 6–7; the boundary $(4, 3)$ regime requires $B \approx 8$; the 50% boundary $(4, 4)$ and polluted $(3, 5)$ regimes do not converge even at high $B$.

## 5.2 Hub Vulnerability Paradox

The predictive value of centrality depended strongly on information quality (Figure 3). In clean regimes ($\geq 70\%$ truth), degree centrality strongly predicted success ($\rho$ up to 0.9, Cohen's $d > 2$), and betweenness, closeness, and eigenvector were also positively associated. In the boundary regime $(4, 3)$ at 57.1% truth, effects were weaker but remained positive ($\rho \approx 0.4$–$0.5$).

At the 50% boundary $(4, 4)$, correlations collapsed ($\rho \approx 0.2$ or less for all four measures), indicating that structural position no longer provided systematic advantage. In polluted regimes ($\leq 40\%$ truth), the direction reversed: degree centrality became negatively associated with success ($\rho \approx -0.5$), closeness and eigenvector dropped to about $-0.6$, and betweenness showed weaker but still negative values ($\rho \approx -0.3$).

This inversion defines the *hub vulnerability paradox*: the same structural advantage that accelerated recovery in clean environments amplified misinformation in polluted ones. Bootstrap 95% CIs for $\rho$ (5,000 resamples within run, pooled by regime) confirmed that the sign reversal was robust. For example, in the polluted $(3, 5)$ regime, the 95% CI for degree centrality was $[-0.55, -0.44]$.

## 5.3 Convergence Timing

When convergence occurred, it happened within a few rounds. Across successful runs, the median convergence time was one round and no run required more than four. Using an alternative definition of convergence ($\geq 90\%$ of agents reconstructing the KB), the window remained 1–3 rounds, confirming that the speed is not an artifact of the first-valid-proposer rule.

Runs that failed to converge stalled quickly: in 36 of 50 non-convergent cases (72%), agents stopped acquiring new facts well before the 30-round cap. Supplementary experiments with very low bandwidth ($B = 1, 2$) and very high bandwidth ($B = 14, 20$) reinforced this pattern: low-$B$ never

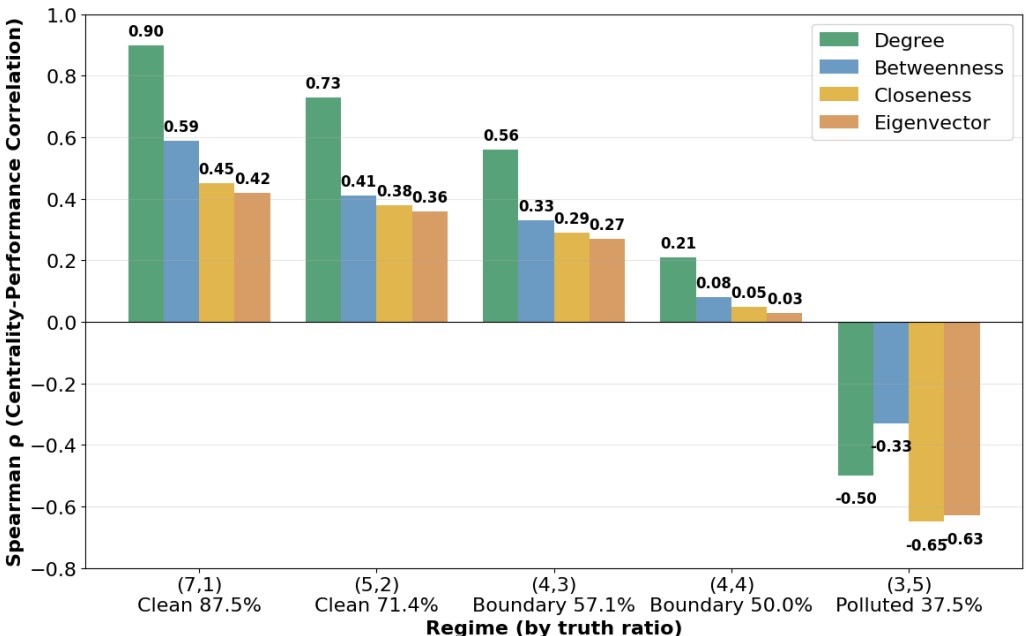

Figure 3: **Hub vulnerability paradox: centrality correlations reverse with information quality.** Spearman correlations ($\rho$) between centrality measures and agent performance across information regimes. All correlations $p < 0.001$ by permutation test; bootstrap 95% CIs confirm sign robustness.

converged due to insufficient transmission, while high-$B$ still failed in boundary and polluted regimes, showing that extra capacity cannot offset poor information quality.

## 6  Discussion

Our experiments revealed three consistent findings. First, collective recovery depends on bandwidth thresholds: once $B$ crosses a regime-specific critical value, networks converge almost immediately, whereas below it recovery never occurs. Second, information quality determines the role of centrality. In clean environments, hubs act as accelerators of recovery, but in polluted environments they amplify falsehood, producing a *hub vulnerability paradox*. At the 50% boundary, structural effects collapse and networks fail to converge. Third, convergence is discontinuous: when recovery occurs, it does so within a few rounds; otherwise the system stalls. These results show that belief change under misinformation is not gradual but threshold-driven, resembling phase transitions in a qualitative sense rather than formal critical phenomena.

These findings highlight structural weaknesses in information systems. A well-connected hub is not always beneficial; its value depends entirely on the reliability of the information it transmits. When inputs are clean, hubs accelerate diffusion; when inputs are noisy, they spread error more efficiently than peripheral nodes. The collapse near the 50% boundary demonstrates how small shifts in the signal-to-noise ratio can trigger large systemic differences in whether truth is recovered or lost. This tipping-point behavior extends ideas from network diffusion and social learning, emphasizing that resilience is conditional, not structural.

As noted in Methods, we also ran small illustrative demonstrations with LLaMA 3.0 agents to link the symbolic results to emerging AI multi-agent systems. In these runs (50–100 agents on subnetworks, up to 10 rounds), each LLaMA process represented a node: it received its current assertions and neighbor messages as input, generated an updated belief set in natural language, and returned a shortlist of top-scored items. The agents typically repeated and reinforced neighbor information rather than applying consistency checks, and sometimes deferred on contradictions, qualitatively mirroring the redundancy heuristic. These demonstrations serve as plausibility checks showing that

symbolic redundancy captures behaviors observable in LLM interaction. They were illustrative only, not statistical evaluations, but they motivate systematic tests of whether the hub vulnerability paradox extends to large-scale multi-agent LLM collectives.

Several limitations qualify these findings. Agents updated beliefs only through redundancy, without credibility weighting or inference, which makes recovery harder and thus results a conservative lower bound. Assertions were treated as independent, networks were static, and communication followed a uniform broadcast protocol. Robustness checks—longer horizons, random tie-breaking, and stricter convergence definitions—produced qualitatively similar outcomes, indicating that thresholds and the paradox are not artifacts of design choices. Future work should extend the model to richer heuristics, dynamic or adaptive networks, and structured knowledge bases, and evaluate the phenomena across multiple synthetic and empirical networks. At the AI side, larger controlled experiments with LLM collectives would clarify whether the same vulnerabilities emerge when models interact at scale.

## 7 Conclusion

This study examined how networks recover ground truth when initial knowledge is incomplete and contaminated by misinformation. Using symbolic simulations on a large collaboration network, we showed that recovery depends jointly on information quality, communication bandwidth, and network structure. Systems either converge quickly to the correct knowledge base or fail entirely, indicating that truth recovery is a threshold process rather than a gradual diffusion.

Within this framework, we identified three informational regimes. In clean environments, hubs accelerate recovery; at the 50% boundary, structural advantages collapse; and in polluted environments, hubs amplify falsehood, producing a *hub vulnerability paradox*. This paradox highlights how the same connectivity that supports efficient truth recovery can create fragility when misinformation dominates.

These results connect to concepts of percolation and tipping points in network science, showing that resilience depends on both topology and the quality of transmitted information. For social networks, interventions such as decentralization or hub bandwidth limits may reduce vulnerability. For multi-agent LLM systems, our small demonstrations suggest similar risks, but systematic evaluation is a task for future work. More broadly, robustness arises not from structure alone but from the joint alignment of topology, communication capacity, and information quality.

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

## Responsible AI Statement

This research was produced with substantial involvement of AI systems. The AI was responsible for producing most of the simulation code, analyzing outputs, and drafting large portions of the manuscript. Human authors verified correctness of the code and results, and edited the text for precision and consistency.

The use of AI accelerated experimentation and writing, making it possible to explore multiple design choices and present results more efficiently. At the same time, AI systems occasionally introduced details that were not implemented in code or overstated the strength of certain findings, requiring human correction.

## Reproducibility Statement

We have taken steps to ensure reproducibility of our results. The dataset (ca-GrQc collaboration network) is publicly available from the SNAP repository. Simulation parameters, including the number of propositions, truth/false fact initialization ratios, bandwidth values, and random seeds, are fully described in the paper. Each regime–bandwidth configuration was replicated three times, yielding 93 runs and over 150,000 agent-level records. All update rules and contradiction resolution rules are specified mathematically.

While anonymized code cannot be released at submission time, we will provide complete simulation scripts and configuration files upon acceptance.

## Agents4Science AI Involvement Checklist

1. **Hypothesis development**
   Answer: [B]
   Explanation: AI was used to suggest possible framings and directions, but the authors generated the central research question and finalized the research hypotheses. AI input was supportive rather than leading.

2. **Experimental design and implementation**
   Answer: [D]
   Explanation: The simulation code, parameter settings, and most of the implementation were produced by AI systems. Human involvement was limited to debugging, validation, and ensuring alignment with the research objectives.

3. **Analysis of data and interpretation of results**
   Answer: [C]
   Explanation: AI produced the bulk of descriptive summaries, preliminary interpretations,

and draft correlations. The authors verified the outputs, corrected inaccuracies, and provided the final interpretations.

4. **Writing**

Answer: [C]

Explanation: AI generated the majority of text across sections. The authors revised drafts for accuracy, removed redundancy, and ensured the writing was precise and consistent with the results.

5. **Observed AI Limitations**

Description: While AI assistance accelerated drafting and prototyping, it also introduced recurring limitations. Generated text was often verbose, repetitive, or imprecise, and required substantial human editing to maintain clarity and academic tone. Code suggestions sometimes ignored model assumptions. In several instances, the AI produced fabricated datapoints or misleading visualizations when prompted to generate analysis or figures. The AI also tended to overstate the robustness of results without verifying statistical validity. These issues made continuous human oversight essential to ensure methodological soundness, validate outputs against raw data, and refine the interpretation of findings. Human authors were responsible for checking code execution, re-running experiments, verifying figures, and editing the narrative.

## Agents4Science Paper Checklist

